# Towards Low-Cost Hyperspectral Single-Pixel Imaging for Plant Phenotyping

**DOI:** 10.3390/s20041132

**Published:** 2020-02-19

**Authors:** Mathieu Ribes, Gaspard Russias, Denis Tregoat, Antoine Fournier

**Affiliations:** 1Department of Crop Phenotyping, Arvalis-Institut du Végétal, 45 voie Romaine, Ouzouer le Marché, 41240 Beauce-la-Romaine, France; 2Department of Biophotonics, Photonics Bretagne, 4 rue Louis de Broglie, 22300 Lannion, France; grussias@photonics-bretagne.com (G.R.); dtregoat@photonics-bretagne.com (D.T.)

**Keywords:** plant phenotyping, proximal sensing, hyperspectral imaging, single pixel imaging, Fourier patterns

## Abstract

Hyperspectral imaging techniques have been expanding considerably in recent years. The cost of current solutions is decreasing, but these high-end technologies are not yet available for moderate to low-cost outdoor and indoor applications. We have used some of the latest compressive sensing methods with a single-pixel imaging setup. Projected patterns were generated on Fourier basis, which is well-known for its properties and reduction of acquisition and calculation times. A low-cost, moderate-flow prototype was developed and studied in the laboratory, which has made it possible to obtain metrologically validated reflectance measurements using a minimal computational workload. From these measurements, it was possible to discriminate plant species from the rest of a scene and to identify biologically contrasted areas within a leaf. This prototype gives access to easy-to-use phenotyping and teaching tools at very low-cost.

## 1. Introduction

Global agriculture production is challenged by increasing demands from rising population and a changing climate, which may be alleviated through the development of improved crop cultivars. Modern techniques for crop improvement rely on both DNA sequencing (called genotyping) and the accurate quantification of plant traits (called phenotyping) to identify genes and germplasms of interest. With rapid advances in DNA sequencing technologies, plant phenotyping now represents a bottleneck in advancing crop yields. This phenotyping bottleneck [1] is now being tackled by the scientific community with instrumental, infrastructural, method, and model developments to face this major challenge [2]. However, not all of these solutions are affordable and expandable. The importance of such instruments for phenotyping is a key issue to develop widespread cost-efficient technologies [3]. In the matter of sensor development, the use of technical breakthroughs in bioimaging [4] should help to obtain game-changing tools like low-cost hyperspectral imagers.

Hyperspectral imaging is an emerging technique that was initially developed for remote sensing, but has since found application in different fields such as security and defense, food inspection, biology, pharmaceuticals, and astronomy [5], since it is a non-destructive technique with low cost per analysis. Hyperspectral imaging consists of appending a spectral dimension to a two-dimensional imager. Thus, each pixel will contain information of reflectance over a few hundred spectral channels, resulting in a three-dimensional data object called a hypercube. Different techniques have been proposed, depending on the applications [6], but hyperspectral imaging suffers from trade-offs between the spatial resolution, spectral resolution, light collection, and acquisition rate. Some designs enable the constraint of several of these parameters simultaneously, [7] among them, compressive sensing seems to be a promising method [8].

In the agricultural area, hyperspectral reflectance imaging was primarily used for landscape spatial and seasonal dynamics. Its application to proximal plant phenotyping at the decimeter to meter sensing range makes it possible to infer chemical compositions and internal physical structures based on the spectral signatures of the sample’s reflectance [9]. Several portable hyperspectral sensors have been evaluated for estimating photosynthetic capacities and quenching [10,11,12], but those academic instruments are still too expensive for furnishing extensive experimental networks.

In this article, we propose the use of single-pixel imaging techniques [13] which rely on one affordable detector instead of an array of detectors. The aim here is to develop a low-cost hyperspectral single pixel imaging (HSPI) prototype for the plant phenotyping community. The focus is set on instrumental development, therefore, experimental validation is carried exclusively in terms of sensor accuracy and performance.

## 2. Materials and Methods

### 2.1. HSPI Principle and Setup

Single-pixel imaging enables one to reproduce a parsimonious under-sampled image of a scene using a single detector and several acquisitions [14]. For each measurement, a specific pattern is projected onto a scene and an associated spectral coefficient is acquired. Mathematically [15], a measurement, m_k_ ∈ ℝ^+^ (photon counts), of a scene, S ∈ ℝ_M×N_, of height M and width N can be expressed as a function of the integration time of the detector, Δt (s), where the power is emitted by the source, ϕ_0_ (photons counts/s), a p_k_ pattern, ∈ ℝ_M×N_, and the dark current of the detector, D (photons/s), such that:

m_k_ = Δt (ϕ_0_ p_k_^t^ S + D),
(1)

It is therefore important to choose the projected pattern basis which is adequate for the targeted application (e.g., Hadamard [16], wavelets [17], Fourier [18], etc.). For this prototype, we propose to use Fourier patterns to reduce both the acquisition and reconstruction times of spectra. Thus, the acquisition of a measurement m_k_ depends on the spatial frequencies to be measured for which two p_k_ patterns (real cosine and imaginary sine) must be projected. Due to conjugate symmetry of the Fourier transform’s properties, the acquisition of a half Fourier spectrum is sufficient to reconstruct it in its entirety. Therefore, the number of Fourier patterns required to project on the scene is equal to the number of pixels required for acquirement.

For the study of vegetal areas, a reflectance imaging device is more appropriate than a transmittance one. In addition, the spatial modulation of light requires high projection rates in order to display every needed pattern in a minimum acquisition time. Digital micromirror devices (DMDs) seem to be the most appropriate choice for our HSPI application. The DMD used in this prototype is part of a commercial projector (Acer PD113) with a modified optical chain for our specific application, in order to extend both its spectral range and the output luminous flux. This projector has simply been connected to a computer using a VGA cable. The spectral range of this prototype covers wavelengths from 400 to 750 nm. The display frequency of this projector is less than 100 Hz, so this prototype has a moderate acquisition rate.

In order to develop a hyperspectral imager, a VIS-NIR spectrometer (Flame, Ocean Insight—Idil Fibres optiques, Lannion, France) for general purpose was used for detection. It consists of a light-diffracting grating on a strip of 2048 charge coupled device (CCD) detectors. We used MATLAB^®^ to interface the prototype. Nevertheless, free or open source libraries allow simple control through a USB cable at low investment with high level languages (e.g., Python or Java). 

The total duration time, ΔT_tot_, for a complete acquisition of a K pixels image depends on both the integration time, Δt, of the spectrometer, the number of scans, n_scan_, needed to average the measured spectra, and the display time, ΔT_pause_, of the images. It is defined as follows:

ΔT_tot_ = K (n_scan_ Δt + ΔT_pause_),
(2)

Integration times of the spectrometer are determined before each complete acquisition depending on the nature of the scene and its distance from the detector to it. It is necessary to determine the best geometrical compromise in order to minimize acquisition times by measuring sufficient optical fluxes and also to collect a detection field adapted to the scene dimensions.

At the end of an HSPI acquisition, a half Fourier spectra hypercube is obtained, i.e., a half spectra cube associated to each wavelength sampled by the spectrometer. Fourier spectra hypercubes are reconstructed by a conjugate symmetry and then converted to image hypercubes by two-dimensional inverse Fourier transformation. A reflectance standard must be placed within the acquired scene in order to normalize the reflectance measurements. The resulting hypercubes were finally analyzed here using MATLAB^®^, Python, and Gerbil [19].

A schematic diagram of all the elements of the HSPI assembly is shown in Figure 1.

### 2.2. Instrumental Performances and Improvements

On the one hand, this imager inherits the optical resolution of the spectrometer (0.035 nm) and its spectral sensitivity (6.8 nm, full width at half maximum). On the other hand, the angular resolution of the imager depends only on the projector resolution (1280 × 1024 pixels) and aperture (41° × 33°). The effective spatial resolution in the sampled plane is driven by the distance to the scene.

Several additional effects must be considered when imaging. Firstly, the optical chain’s response is non-linear. This effect has the consequence of stretching the values of the Fourier coefficients in the areas where the behavior of the curve is less linear. To solve this problem, Fourier patterns are designed by setting the grayscale levels in a linear response regime. Then, a ground glass is placed in front of the collection fiber in order to homogenize its angular sensitivity and thus to limit light fall-off effects on images. Finally, since the Fourier patterns are sinusoids, their values oscillate between ±1, but digital images are defined for positive values only. To solve this problem, we used a pattern splitting technique [20]. For each theoretical pattern, we associate one pattern with its positive values and one with its negative values, taken as absolute values. The acquisition then comes back to the subtraction of the two projections of the associated patterns.

### 2.3. Reflectance Reference Measurements

To evaluate the accuracy of the reflectance measurement by the proposed setup, we obtained the reflectance truth with a conventional point measurement at a distance of few centimeters. These measurements were performed with a bifurcated optical fiber (QR400-7-VIS-NIR, Ocean Insight—Idil Fibres optiques, Lannion, France) connected to a fibered light source (HL-2000-HP, Ocean Insight—Idil Fibres optiques, Lannion, France) and the same spectrometer in order to measure annulus conical reflectance factors. This quantity is comparable to the conical-conical reflectance factor (CCRF) obtained with HSPI sensors [21].

### 2.4. Biophysical Index and Biological Material

A simple way to study hypercubes and discriminate the vigor of plant material within a scene from a hypercube is via the normalized difference vegetation index (NDVI) [22]. This index quantifies the contrast between the near-infrared domain and red wavelengths. Here, we used wavelengths at the end of the detection level at 750 nm to sample the values in the near-infrared reflectance rise of the plants and at 630 nm for the red wavelengths.

The biological material used here was leaves of hortensia (*Hydrangea macrophylla* L.). The leaves of hortensia were collected during summer 2019 from Photonics Park (Lannion, France). A homogeneous green leaf and a highly stressed leaf were respectively selected for the experiments. The continuity of the water column was ensured by cutting petioles under water and then by maintaining them in water reservoirs during the acquisitions. Reflectance point measurements were carried out prior to each image acquisition. These measurements were made in reflectance using the same calibration standard present in the scene. 

## 3. Results

In order to illustrate the performances of the HSPI prototype presented above, we provide the results of two experiments. The first one aims to discriminate between a hydrangea leaf and its environment and the second one to examine different health states within the same leaf. 

### 3.1. Discrimination of a Leaf within a Scene

For this first experiment, the scene consists of a hydrangea leaf, a photocopy, and a reflectance calibration standard, stuck on a black paper background. This is visible in Figure 2a. In addition, pieces of white adhesive tape protrude from both leaves to enhance the impact of backscatter transmitted light. The image hypercube was reconstructed with a spatial resolution of 81 × 81 pixels, a spectrometer integration time of 190 ms, with 5 averages per spectrum, and a pattern display rate of 200 ms. The duration time of this acquisition was about 4 h and the reconstructed image, in artificial colors, using Gerbil and the spectra associated with each element of the scene, is presented in Figure 2. The data of this hypercube and a video are available in the Appendix A.

The HSPI prototype allows one to reproduce spectral shapes corresponding to those measured locally for each element of the scene. Thanks to the spectral shapes, it is possible to distinguish the plant leaf from the other elements using images at different wavelengths and NDVI mapping, as represented in Figure 3.

### 3.2. Detection of Different Health States on a Same Leaf

The sample was selected in such a way that it was possible to examine the spectral signatures of different health states within the same leaf. A photograph of the scene and its associated HSPI are shown in Figure 4, as well as the reflectance spectra, measured on selected areas representative of the different states of the health of the leaf.

This image was reconstructed with a resolution of 101 × 101 pixels, a spectrometer integration time of 58 ms, with 5 averages per spectrum, and a pattern display rate of 100 ms. The duration time of this acquisition was just over 2 h. Data of this hypercube are available in Appendix A. The HSPI prototype thus makes it possible to correctly discriminate between different health states within a strongly stressed leaf. A line plot of a transection of the NDVI mapping allows the analysis of each area, which is presented in Figure 5.

## 4. Discussion

We have established a moderate flow and low-cost HSPI prototype for the phenotyping of vegetal species. This prototype uses a modified commercial projector and a general-purpose spectrometer to allow hypercube acquisitions over the 400–750 nm spectral range. It permits one to access unprecedented working distances and scene sizes with desk-grade material and is easily reproducible. This prototype should increase access to hypercube data for the plant science and phenotyping community.

The maximum resolution expected at working distances of 30 cm (Figure 2b) and 15 cm (Figure 4b) is about 0.16 and 0.08 mm per pixel, respectively. Practically, the pattern basis employed affects the accessible resolution of the reconstructed images. Here, the effective resolution depends on the requested number of pixels. As stated above, the pixel number to sample in Fourier space equals the number of patterns for projection. Then, the image resolution can be set for each acquisition considering the duration/resolution trade-off. In Figure 2b and Figure 4b, the effective size of each pixel is respectively 2.6 mm and 1 mm.

Figure 3a shows that it is possible to observe the adhesive tape under the leaf in the near-infrared wavelengths while the photocopy does not transmit at any wavelength. Indeed, in the near-infrared region, the leaf transmits almost as much light as it reflects, while its transmission is lower between 400 and 700 nm [7].

Among various spectral approaches, the NDVI has been used for a long time to discriminate the vigor of vegetal species. As shown in Figure 5a, an NDVI mapping of the reconstructed image of a leaf sample allows one to identify areas presenting different health states. In Figure 5b, a transection of this index over the leaf highlights the detectability of necrotic, chlorotic, and healthy areas within the leaf.

Several instrumental effects can be deduced from our experimental results. Firstly, a vignetting effect is present for the reconstructed images. It can be attributed to the light fall-off of the optical fiber collection. If the calibration reflectance standard is too far from the collection optical axis, this effect could significantly affect the level of the resulting absolute reflectance values. Then, in Figure 2d, the noisier measurement observed at 750 nm is attributed to the lower illumination from the projector in this spectral range. This widening of the relative noise in the near infrared range can also be seen in each spectrum acquired. In addition, despite the optimization of various elements in the optical chain, the spectral range remains limited by the reflectivity of the projector’s DMD, which cuts the measurable spectral range at 750 nm. Moreover, an additional peak is systematically encountered near 550 nm which can be explained by the thermal drift of the projection lens during the acquisition. A spectral fluctuation has been identified and seems to be positively correlated to the gloss of the imaged objects. The specular area may encounter a drift of its reflectance response during the acquisition. A reduction of the acquisition time and a better control of the temperature surrounding the setup could fix this issue. Finally, a more advanced way to create Fourier patterns would be to fit the whole range of the instrumental chain response and invert it when creating the patterns. This will increase the contrast instead of limiting the illumination power to the span of the hardware linear response. Spectra were smoothed here using a moving average filter with a span of 5 pixels to lower the apparent noise from the charge couple device. 

Figure 2b shows a 101 × 101 pixels measurement obtained over 4 h. The acquisition time can be reduced by three complementary optimizations, namely algorithmic, electronical, and optical optimizations. The algorithmic improvement of compressive sensing acquisitions is an active field of studies [15,23,24]. Fourier patterns are used in the study for the sake of simplicity but could be benchmarked against the basis of other patterns and reconstruction strategies. Electronic improvement consists of increasing the display rate and managing the synchronization of all devices. The use of a suitable and commercially available DMD will allow one to increase the display rate up to the kilohertz level, provided there is an electronic-based control to overcome the limitation that is intrinsic to synchronization based solely on the software layer. Optical improvement aims to increase the power in order to reduce the integration time of the spectrometer. The HSPI setup developed here uses a standard video projector lamp. The use of a more powerful light source would not only make it possible to measure larger optical fluxes but also to reduce the integration time of the spectrometer. Faster and more sensitive spectrometers could also enhance the overall detectability and measurement rate, but with the risk of drifting the setup cost out of the affordable region. With an ideal configuration, reaching a sampling rate in the kilohertz range, the measurement time of a hyperspectral image of 81 × 81 pixels should take around 10 s, as compared to 4 h using the initial prototype.

The current storing method of the hypercube is capable of optimizing memory. Indeed, only the half Fourier spectra are stored, and a set of scripts is used to reconstruct images and carry out analyses based on them. Moreover, it is well-known that two-dimensional inverse fast Fourier transform algorithms are more efficient when the image dimensions are powers of 2. Fourier spectra sampling imposes an odd image format because of the center frequency. This is because a half Fourier spectrum contains a central column that is not duplicated by conjugate symmetry. An improvement would be to get closer to this type of image size by favoring consecutive odd sizing.

The price of this protype, based on laboratory-grade equipment, reaches 4000 €. The spectrometer could be transparently exchanged with even more accessible detector parts, reaching an expected setup price lower than 2000 €.

The most liming factor for outdoor measurements with an active sensor is solar competition, where the optical flux source needs to be more powerful and differentiated from the solar irradiance. In order to have a device working in the field, the optical source should be pulsed for gated measurements.

Data have also been acquired for wheat and maize and will be the subject of further studies to work on practical agricultural applications.

## 5. Conclusions

This paper has introduced an original way to use a Fourier-based patterning strategy for low-cost hyperspectral single pixel imaging. The use of a commercial projector and general-purpose spectrometer allow easy and agile command and control of the setup. With the help of this prototype, hypercubes that allow discrimination of a leaf from a scene or the analysis of the state of health of a leaf have been recorded at moderate distance of 30 cm. This preliminary work is the first step towards high-speed phenotyping sensors. Furthermore, an open implementation of such an original and low-cost setup will enhance the diffusion of these techniques towards cost-efficient wide phenotyping networks. Better time performances will be a key feature for potential integration on conveyer-based imaging chambers of indoor phenotyping platforms. The diversity of photonics skills requested for such a setup of simple devices makes it a good framework for educational purposes or for phenotyping teams to start handling hyperspectral images at low-cost.

## Figures and Tables

**Figure 1 sensors-20-01132-f001:**
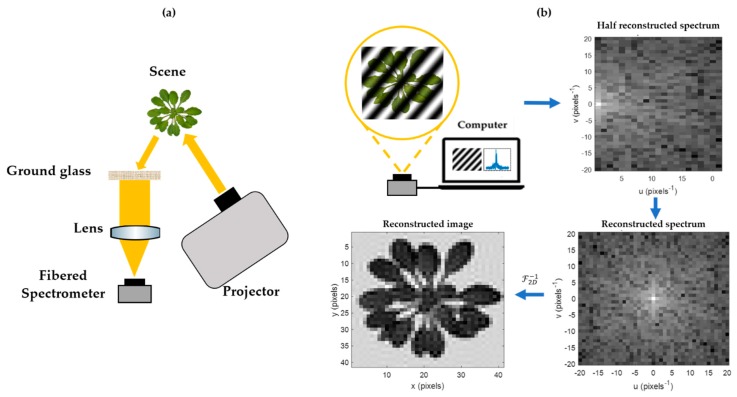
Hyperspectral single pixel imaging (HSPI) acquisition and reconstruction methods in the reflectance and experimental setup. (**a**) Experimental HSPI setup, consisting of a commercial video projector displaying Fourier patterns onto a scene and a lens focusing the reflected optical flow on a detection part. Detection was performed by a general-purpose fibered spectrometer for spectrally sampling acquisitions. (**b**) Simulation of a Fourier-based single pixel imaging acquisition with a resolution of 41 × 41 pixels.

**Figure 2 sensors-20-01132-f002:**
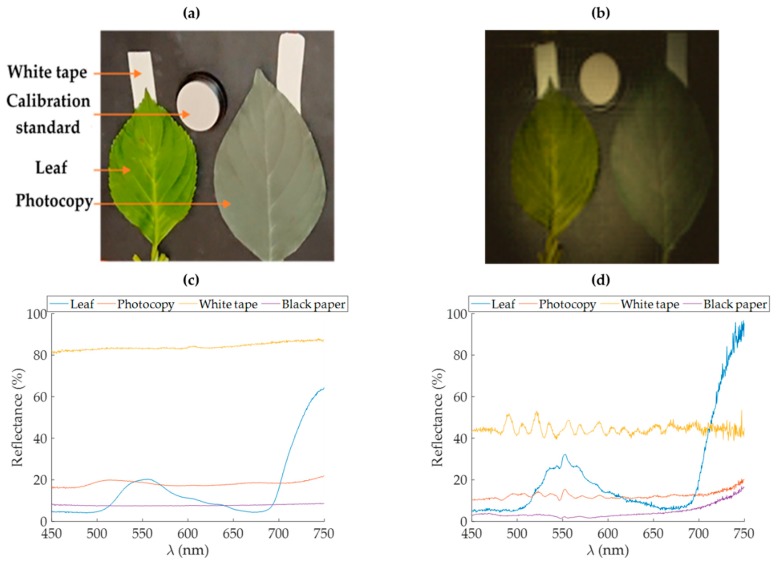
Discrimination of a leaf within a scene. (**a**) Photography of the scene. (**b**) The 81 × 81 pixels false color image reconstructed in HSPI using Gerbil. Comparison of the spectra of each element established by point measurements (**c**) and from the HSPI hypercube acquired (**d**). In the reconstructed picture represented in (**b**), a vignetting effect is visible. It affects the measured reflectance values and is strongly apparent in the white tape’s signature plotted in (**d**). Curves are smoothed with a moving average filter of a 5-pixel span.

**Figure 3 sensors-20-01132-f003:**
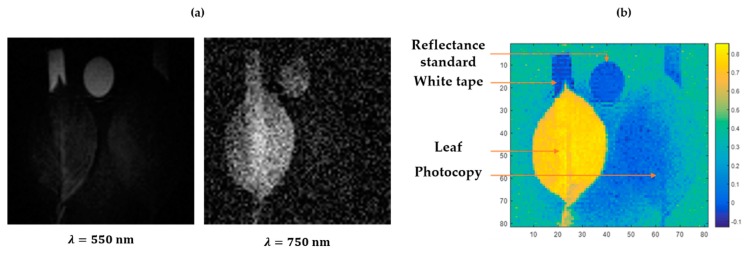
Discrimination of a leaf within a scene. (**a**) Images reconstructed at 550 and 750 nm. (**b**) NDVI mapping of the reconstructed image.

**Figure 4 sensors-20-01132-f004:**
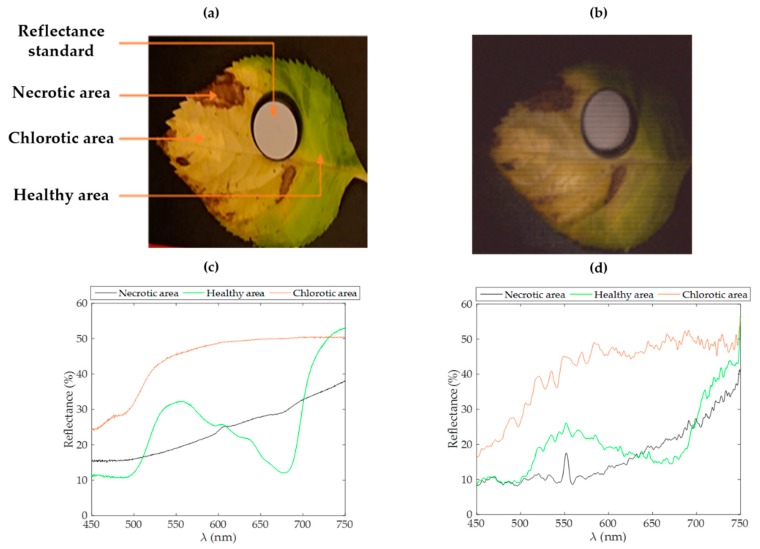
Detection of different health states within a hydrangea leaf. (**a**) Photography of the scene. (**b**) A 101 × 101 pixels false color image reconstructed in HSPI using Gerbil. Comparison of each element’s spectra, as established by point measurements (**c**) and from the HSPI hypercube (**d**). Curves have been smoothed with a moving average filter with a span of 5 pixels.

**Figure 5 sensors-20-01132-f005:**
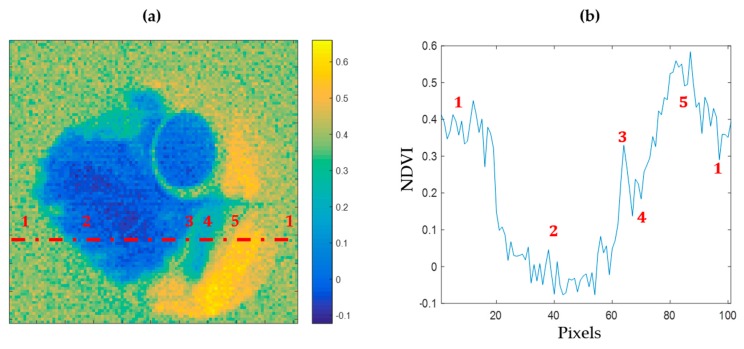
Detection of different health states within a hydrangea leaf. (**a**) NDVI mapping of the reconstructed image. (**b**) Analysis of different health states from the red dotted line of NDVI map.

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
