# Peer review of "Towards Low-Cost Hyperspectral Single-Pixel Imaging for Plant Phenotyping"

_sensors, 2020, doi:10.3390/s20041132_

Round 1
Reviewer 1 Report
The authors are discussing an impressive way to capture single-pixel hyperspectral imaging in agronomy. The main focus is on instrumental design and technology. However, there are some points to consider before moving on.
Introduction:
1. This section needs improvements. Although it discusses the main objective of the study in short, it should include other literatures and it should synthesize the results from several papers to produce argument about the technology that is being developed.
Materials & methods and results:
2. The authors might think about making the coding available in Github or similar platforms.
3. To be able to conclude results, a standard experiment needs to be designed and developed.
3. After data collection using the new tool, proper statistical analysis should be performed in order to infer the results.
Discussion:
4. This part discusses the results quite well. However, the experiment is not designed in a way to be able to infer such results yet. This section needs to be updated after conducting the proper experiment.
This work is a much needed research in agronomy, and it is discussing an state-of-art methodology. I highly encourage the authors to conduct an experiment with the tool and submit this paper again.
Author Response
Dear reviewer,
We thank you for your kind suggestions which have helped us to improve significantly our article. Please find attached elements of answers to your questions.
Kind regards,
Mathieu Ribes on behalf of all authors

Reviewer 2 Report
Topic presented in this paper is interesting, however, which may be partly due I am not an expert in this field, there are some things unclear for me. Furthermore, the structure of the paper should be improved.
Firstly, the introduction part is very short. After reading that I don't feel that this device/technique is really important. Could you please extend the introduction part with, e.g. applications or a little bit more description of phenotyping and hyperspectral imaging. I suggest to make it more evident in the introduction that this study is really needed.
Also, methodology section is written in a bit chaotic way. I suggest to put maybe a graphical workflow with all the steps. There are some methods that you used missing in the methodology, e.g. NDVI, and the fact that you examine healthy and unhealthy leaves - you first mention it in the results section!
I have also some doubts about using NDVI in this case. As I know NDVI is a popular technique in mapping vegetation with e.g. satellite images and it uses near-infrared and visible red part of the spectrum. However, as you described, examined waves in your study range from 400 to 750 nm. In my field, that wavelengths are considered as visible waves or red-edge, not infrared. Could you please explain why you consider these waves as infrared?
My another doubt is indeed related to infrared part of the spectrum. Why your device does not measure reflectance in longer wavelengths, namely NIR or SWIR? I guess these are the parts of the spectrum most commonly used in monitoring e.g. vegetation health.
Regarding the results, I think the time of the measurement is rather long (2 or 4 hours). Then are there any perspectives to make it shorter in order to apply that, for example, in the field work? And again, in the discussion part there are some missing information and it is very short. How to apply that study? Why is it important? How the results look in comparison to other similar studies?
Author Response

(The authors gave the same response as above.)

Reviewer 3 Report
I suggest to complet the materials and methods part with some data included in the discussion.
You should discuss more the concerns about the time of data acquisition and the feasability to make the extensability to field phenotyping.

Author Response

(The authors gave the same response as above.)

Round 2
Reviewer 2 Report
The paper has now better structure and the importance of this topic is explained more clearly. All my comments from my review were properly adressed.